# *Trichoderma atroviride* Enhances *Impatiens walleriana* Hook. f Growth and Flowering in Different Growing Media

**DOI:** 10.3390/plants13050583

**Published:** 2024-02-21

**Authors:** Silvia Traversari, Mariateresa Cardarelli, Massimo Brambilla, Maurizio Cutini, Gianluca Burchi, Marco Fedrizzi, Daniele Massa, Alessandro Orlandini, Sonia Cacini

**Affiliations:** 1Research Institute on Terrestrial Ecosystems (IRET), National Research Council (CNR), Via Giuseppe Moruzzi 1, 56124 Pisa, Italy; silvia.traversari@cnr.it; 2National Biodiversity Future Center (NBFC), 90133 Palermo, Italy; 3Department of Agriculture and Forest Sciences, University of Tuscia, Via S. Camillo De Lellis snc, 01100 Viterbo, Italy; 4Research Centre for Engineering and Agro-Food Processing, Council for Agricultural Research and Economics (CREA), Via Milano 43, 24027 Treviglio, Bergamo, Italy; massimo.brambilla@crea.gov.it (M.B.); maurizio.cutini@crea.gov.it (M.C.); 5Research Centre for Vegetables and Ornamental Crops, Council for Agricultural Research and Economics (CREA), Via dei Fiori 8, 51017 Pescia, Pistoia, Italy; gianluca.burchi@crea.gov.it (G.B.); daniele.massa@crea.gov.it (D.M.); 6Research Centre for Engineering and Agro-Food Processing, Council for Agricultural Research and Economics (CREA), Via della Pascolare 16, 00015 Monterotondo Scalo, Rome, Italy; marco.fedrizzi@crea.gov.it (M.F.); alessandro.orlandini@crea.gov.it (A.O.); 7Research Centre for Agriculture and Environment, Council for Agricultural Research and Economics (CREA), Via di Lanciola 12/a, 50125 Firenze, Italy

**Keywords:** bedding plants, biostimulant effect, chlorophyll content, coconut coir dust, peat, *Trichoderma* spp.

## Abstract

*Trichoderma* spp. are widely reported to regulate plant growth by improving nutrient uptake, photosynthesis, and abiotic stress tolerance. However, their possible application for bedding plants is little explored, especially when comparing different growing media. Considering that coconut coir dust is finding broader application in the ornamental plants sector as a peat substitute, this work was aimed to test the combination of *Trichoderma atroviride* AT10 and coconut coir dust on *Impatiens walleriana* plants. Four treatments were tested as a mix of: (i) two growing media (70:30), peat:perlite or coconut coir dust:perlite; and (ii) the absence or presence of a *T. atroviride* treatment. At the end of the production cycle, the biomass and ornamental parameters, leaf pigments, nutrient content of the plant tissues, and *Trichoderma* abundance were assessed. The results revealed that *T. atroviride* can readily colonize coir, and the same positive effects of inoculum were found in plants grown on both substrates. The biostimulant effect of *T. atroviride* was observed as an increase in the aboveground biomass, number and weight of flowers, pigments and nutrient concentration, thereby improving the commercial quality of *I. walleriana*. Thus, *T. atroviride* has shown its potential in making bedding plant cultivation more sustainable and improving the yield and aesthetic parameters of plants grown on peat and coconut coir dust substrates.

## 1. Introduction

Bedding and garden plants represent an important market share of the floriculture industry as they are considered space-saving products, especially in some countries, such as the United States [1,2], where they accounted for 47% of the wholesale in 2018, considering the overall market of floricultural products [3]. Looking at the European market, the wholesale was worth 2.2 billion in 2019, with the Netherlands, France, Italy, Germany, Spain, and Poland achieving a 27% quota of production destined for the Extra-EU export market [4]. Bedding plants are a typical soilless culture production that faces environmental and economic challenges, such as the use of more sustainable crop protection strategies and raw materials, including growing media. Indeed, soilless cultures are high-consuming systems that are increasingly asked to reduce inputs, including water, energy, agrochemicals, and the production of waste materials, such as plastics and spent growing media, although ensuring high product quality standards [5,6]. In this regard, molecules and microorganisms with a biostimulant function may play a crucial role in enhancing quality, together with reducing the use of phytochemicals, even if the spread of organic substrates based on other materials than peat requires improving the knowledge of possible interactions.

Coir has become one of the most commonly used peat alternative growing media constituents worldwide, at least in the last two decades [7,8,9], despite some emerging criticisms about its sustainability due to the environmental impact related to composting processes and shipping, as well as the social issues such as community infrastructure problems and human and labor rights respect [7,10]. It is a renewable material with optimal well-known physical–chemical properties [11,12], which allow the successful production of many soilless crops, such as bedding plants, including *Impatiens* spp., used both as a stand-alone substrate or in mixtures, starting from germination and seedling growth [13,14,15]. Coir, as organic matter, has higher microbiological activity compared to peat [16,17], and it has a recognized natural content of plant growth-promoting bacteria (PGPB), such as phosphate-solubilizing bacteria and indole-3-acetic acid-producing bacteria [18,19].

With the implementation of sustainable agronomic techniques, beneficial soil microorganisms are increasingly adopted to enhance the plant growth rate and quality [20,21]. *Trichoderma* spp. are ubiquitous free-living fungi (*Hypocreaceae* family), occurring widely in all soil types and representing major components of their mycoflora [22]. *Trichoderma* fungi can live in soil stress conditions such as salinity, alkalinity, nutrient deficiency, and drought [23]. Many *Trichoderma* species have been extensively studied in open fields and greenhouse cultivation due to their well-known biological control mechanism. Moreover, they are recognized to have a powerful capacity to improve plant growth, physiological traits, nutrient uptake, and yield [24,25,26,27]. Particularly, *T. atroviride* has been shown to increase the biomass of lettuce, zucchini, eggplants, tomato, and radiata pine seedlings [25,28,29,30,31], as well as to promote the root development of *Arabidopsis* seedlings [32,33]. The ability to stimulate plant growth has been positively correlated with the production of metabolites with hormone activities like indole-3-acetic acid or auxin analogues or the solubilization of nutrients like iron through siderophore secretion in the rhizosphere [34,35]. Consequently, the use of *Trichoderma* spp. as an active ingredient in biofertilizers, biopesticides, bioremediates, and natural resistance stimulants is becoming quite common. Despite this, to the best of our knowledge, there is no scientific information on the use of *Trichoderma* in bedding plant cultivation and little information on its application on ornamental plants in general, particularly when testing different growing media. 

This work was aimed to evaluate the effect of *T. atroviride* addition on a typical bedding plant grown on a coir-based substrate as an alternative to peat growing media. Among the wide range of available herbaceous bedding plants, *Impatiens walleriana* has been chosen because it is much appreciated for its wide range of flower colors and long flower display, as well as for its attractive leaves.

## 2. Results

### 2.1. Biomass Measures

The application of *T. atroviride* statistically increased the aboveground biomass of *I. walleriana* plants, while the substrate type did not affect these parameters (Table 1). The plant shoot fresh (FW) and dry weight (DW) were positively influenced by *T. atroviride* treatment in both substrates (+26 g FW plant^−1^ and +0.8 g DW plant^−1^ in treated plants, a 24 and 16% increment, respectively, considering the average values on peat- and coir-based substrates). A positive effect was also found in flowers showing a higher FW and number, which resulted in 34 and 47% increases, respectively, considering the average values on peat- and coir-based substrates. Flower DW was unaffected by fungal treatment and substrate (Appendix A). By measuring the number of flowers produced during the growing cycle, the *T. atroviride* presence resulted in significant differences at the end of the trial but not at the first and second flower harvests (Figure 1). The positive effect on flowering is also displayed in Figure 2. The same trend was also observed in the leaf area, for which the treated plants had a 23% increase, considering the average values on peat- and coir-based substrates (Table 1). The specific leaf area (SLA) was not influenced by the treatment and substrate type (Appendix A).

The substrate type influenced the root biomass, while the *T. atroviride* treatment did not improve this parameter (Table 1). Specifically, the *I. walleriana* plants had a statistically lower root biomass in coir than in peat (−15% considering the average values of not treated and treated plants).

### 2.2. Mineral Elements

*T. atroviride* treatment in the shoots influenced almost all the mineral elements, especially in the coir substrate (Table 2). The shoot total Kjeldahl nitrogen (TKN) was influenced by the interaction between the treatment and substrate, and the highest value was found in the not treated plants grown on the coir-based substrate. Even the shoot Ca was influenced by the interaction between the treatment and substrate, and specifically, it was higher in the treated plants than in the not treated ones in the coir-based substrate. Differently, P-PO_4_ and K increased with *T. atroviride* (+7 and 9%, considering the average values on peat- and coir-based substrates). In contrast, Mg was only influenced by the substrate type and was lower in coir than in peat (−9%, considering the average values of the not treated and treated plants). A lower number of statistically significant variations in the mineral elements were found in the roots (Appendix A). Specifically, K was higher in the treated plants (+18%, considering the average values on peat and coir), while Mg was higher in coir than in peat (+11%, considering the average values of the not treated and treated plants).

### 2.3. Leaf Pigments and Trichoderma Analysis

*T. atroviride* treatment promoted the chlorophyll *a* and carotenoid accumulation in the leaves, with values of +9 and 6% compared to the control, respectively, considering the average values on peat- and coir-based substrates (Table 3). Neither the treatment nor substrate showed any effect on chlorophyll *b*. The interaction between the treatment and substrate influenced the greenness index, which was higher for the plants grown on peat and treated with *T. atroviride* (Figure 3). On the coir-based substrate, the treatment with *T. atroviride* did not reveal any difference in the greenness index. The SPAD measures were also higher in the plants treated with *T. atroviride* at the intermediate sampling points (Figure 4).

The fungal presence was higher in the pots of treated plants, resulting in an average of 11–12 *Trichoderma* colony forming unit (CFU) × 10^3^ g substrate^−1^ (+59% of not treated plants, considering the average values on peat- and coir-based substrates, Figure 5).

### 2.4. Principal Component Analysis (PCA)

The PCA (Figure 6) highlighted that 46% of the variance could be explained by the differences between the treated and not treated plants, with the shoot TKN and Mg loading factors mostly identifying the not treated plants. On the contrary, 25% of the experimental variance could be explained mainly by the differences between coir and peat. The peat observations were mostly related to Mg, flower FW, chlorophylls, carotenoids, and root DW; the coir observations were mainly related to the shoot FW and DW, greenness index, and TKN.

## 3. Discussion

The inoculation of the substrate with *Trichoderma* spp. is known to promote growth, flowering, quality, and nutritional status of ornamental plants regardless of the substrate type [36]. An increase in the biomass and leaf area was observed in *I. walleriana* plants in our experimental conditions, as already found in tulip and gladiolus [37,38], polianthes [39], *Lantana camara* [40], begonia [41], sea lavender, cypress, and camellia [42] inoculated with different *Trichoderma* strains. Moreover, a significant effect on flowering performance attributable to the elongation of inflorescences and the development of flowers has been also demonstrated in tulip, polianthes, and begonia [38,39,41]. Inoculating a peat-based growing media with *T. harzianum*, Ousley et al. [43] showed an improvement in the number of flowers in marigold and petunia, while in verbena, both the number and weight of the flowers increased; these results differed from the present work for the applied *Trichoderma* species (*T. harzianum* instead of *T. atroviride*) and its inoculum dose. Despite these promising results, the mechanisms supporting the beneficial effects of *Trichoderma* spp. on plant growth stimulation have not been fully explained. Contreras-Cornejo et al. [44] suggested that the plant growth promotion ability of *T. virens* is mediated by an auxin-dependent mechanism; through in vitro tests the authors confirmed the fungus’s ability to synthetize indole-3-acetic acid (IAA) and some derivatives, leading to greater root development. In the present experiment, however, the root biomass did not increase following the treatment with *T. atroviride*. Alternatively, the higher biomass accumulation can be attributed to improved nutrient availability [45]. *Trichoderma* spp. can in fact facilitate nutrient absorption from the substrate through mineral solubilization and increased element uptake [46]. Indeed, in our experimental conditions, *T. atroviride* increased the leaf macronutrient concentrations (P, K, and Ca), similarly as found by Andrzejak and Janowska [37] in gladiolus and Janowska et al. [47] in freesia, in both cases using a mixture of *T. viride*, *T. harzianum*, and *T. hamatum*.

Most research papers addressing the impact of *Trichoderma* spp. on the content of chloroplast pigments in leaves referred to edible species [48,49,50], while only a few studies investigated the stimulation of photosynthetic pigments on ornamental plants [46]. An increase in chlorophylls was found in *Begonia* × *tuberhybrida* [41] and *Gladiolus hybridus* ‘Advances Red’ [37] soaked in a mixture of spores of *Trichoderma* spp. and planted in peat substrate. According to Andrzejak et al. [41], the chlorophyll accumulation in *Begonia* × *tuberhybrida* was reflected by the greenness index. In our case, a significant accumulation of pigments, mainly chlorophyll *a* and carotenoids, and a higher greenness index, were found in leaves as a consequence of inoculum, therefore influencing the plant photosynthetic capability. In fact, even carotenoids are essential pigments in photosynthesis because they absorb in the blue–green region of the solar spectrum and transfer the absorbed energy to chlorophylls, so expanding the wavelength range of light, which can drive photosynthesis. Indeed, Harman et al. [49] stated that endophytic strains of *Trichoderma* determine an increase in the number of photosynthetic pigments or the expression of genes regulating the biosynthesis of chlorophylls, proteins in the light-harvesting complex, or components of the Calvin cycle.

Regarding the substrate suitability, no differences were observed in either the epigeal biomass production or flowering performance. The suitability of the coco-peat substrate for the application of *Trichoderma* spp. has already been highlighted by Sriram et al. [51], who monitored the *T. harzianum* fungal population density during the time, highlighting a stable population from 28 to 42 days after the inoculum. Our experimental results highlighted the high efficacy of inoculum also in coir alone.

## 4. Materials and Methods

### 4.1. Plant Material and Growing Condition

This experiment was carried out at the Research Centre for Vegetable and Ornamental Crops, Council for Agricultural Research and Economics, in Pescia (PT), Italy (lat. 43°54′ N, long. 10°42′ E, altitude 62 m). The trial was conducted in a greenhouse equipped with benches for soilless cultivation, a capillary fertigation system, and a basal heating system based on coaxial pipes circulating warm water powered by a compressor heat pump [52]. Basal heating was applied during the trial to maintain a temperature of 16 °C at the pot and root level. Cuttings of *Impatiens walleriana* Hook. f ‘Buddha F1 Carmine’ (Azienda Agricola Sentier, Mosnigo di Moriago della Battaglia, Treviso, Italy) were transplanted on 14 February 2019 in 1.2 L pots. Four treatments were applied as a combination of: (i) two growing media, i.e., peat:perlite and coconut coir dust (coir):perlite (both 70:30, *v v*^−1^); and (ii) amended growing media with or without 1 g L^−1^ of a *Trichoderma atroviride*-based inoculant (Tricoten, *Trichoderma atroviride* AT10 at 5 × 10^8^ CFU g^−1^, Atens-Agrotecnologias Naturales SL, Tarragona, Spain). For each treatment, 4 replicates of 20 plants were set up for a total of 320 plants. The growing media pH and electrical conductivity (EC) were determined following the EN 13037/1999 [53] and EN 13038/1999 [54] methods, respectively, which consist of an electrometric determination on a substrate:water (1:5 *v v*^−1^) extract after 30 min shaking of a known volume of sieved wet samples. The physical characteristics (i.e., substrate bulk density, total porosity, available water content, water holding capacity, and air content) were determined as described by De Boodt and Verdonck [55]. The substrate characteristics are reported in Table 4. The trial started with the cutting transplant and ended on 11 April 2019, after 56 d of cultivation. The heating system was set to avoid an air temperature below 5 °C. Fertirrigation was managed according to the weather conditions by supplying a nutrient solution typically used for bedding plant production (i.e., 7.0 mM N-NO_3_, 0.7 mM N-NH_4_, 4.0 mM K, 2.5 mM Ca, 1.0 mM Mg, 1.8 mM S-SO_4_, 30.0 µM Fe, 25.0 µM B, 1.0 µM Cu, 5.0 µM Zn, 10.0 µM Mn, and 1.0 µM Mo) and maintaining a pH of roughly 6.0. During the experiment, the climate condition inside the greenhouse was monitored by a Testo data logger, mod. 175 (Testo SE & Co. KGaA, Titisee-Neustadt, Germany) and the recorded data were: average temperature 16.8 °C and air relative humidity 57.8%.

### 4.2. Plant Biomass Measures

At the end of the experiment, the shoot (stems plus leaves) and flower FW and DW were determined in each biological replicate (i.e., in a bulk sample of 10 plants out of 20 plants constituting a replicate) by drying the samples in a ventilated oven at 60 °C until a constant weight. Before drying, the leaf area was measured in a significant portion of fresh leaves using a WinDIAS Image Analysis System (Delta-T Devices, Cambridge, UK) and used to determine the SLA (cm^2^ g DW^−1^). Flowers were periodically collected during the experiment from all the plants constituting every replicate (three times during the trial, i.e., 40, 49, and 56 d after transplanting, indicated as 1st collection, 2nd collection, and final collection) to obtain the total flower production, expressed as the number, FW, and DW. Roots were sampled at the end of the trial from the same plants used for the aboveground biomass measures (i.e., a bulk sample of 10 plants out of 20 plants constituting a replicate), washed in diluted acetic acid to remove the substrate, and used for the DW measures. The SPAD index was assessed in 4 leaves per plant, in 10 plants per replicate (the same plants used for biomass measures), by a SPAD-502 (Konica Minolta, Inc., Ishikawa-machi, Hachioji-shi, Tokyo, Japan) at the 1st and 2nd collections to monitor the leaf chlorophylls using a not-destructive technique.

### 4.3. Tissue Analyses

At the conclusion of the experiment, fresh leaf disk samples were randomly collected from plants belonging to the same replicate (100 mg FW per replicate in two technical replicates) to assess the concentration of chlorophylls *a* and *b* and total carotenoids (mg g^−1^ FW). The samples underwent a methanol extraction (99%, *v v*^−1^) in darkness at −20 °C for 48 h and were analyzed using a spectrophotometer (Evolution™ 300 UV–Vis Spectrophotometer, Thermo Fisher Scientific Inc., Waltham, MA, USA) to measure the absorbances at 665.2, 652.4, and 470.0 nm, following the method described by Lichtenthaler and Buschmann [56]. The chlorophyll *a*, chlorophyll *b*, and carotenoid concentrations allowed the calculation of the greenness index [56] as follows:Greenness Index = (Chlorophyll *a* + Chlorophyll *b*)/Carotenoids

The root and shoot (i.e., stems plus leaves) dry samples underwent a Kjeldahl nitrogen (TKN) analysis after a phospho-sulfuric acid digestion, as reported by Massa et al. [57]. In addition, the samples (250 mg DW) were digested in a mixture of nitric and perchloric acids (HNO_3_:HClO_4_ 5:2 *v v*^−1^) at 230 °C for 1 h to determine the concentrations (mg g^−1^ DW) of potassium (K), calcium (Ca), magnesium (Mg), and phosphate (P-PO_4_). K, Ca, and Mg were quantified by atomic absorption spectrometry (AA-7000F Flame Atomic Absorption, Shimadzu, Japan), while the P-PO_4_ concentration was determined spectrophotometrically through the molybdenum blue method [58].

### 4.4. Trichoderma atroviride Analysis

Substrate samples were collected from the plant rhizosphere at the first floral sampling and the end of cultivation. Fungus quantification was performed by serial substrate dilutions in a *Trichoderma*-selective agar medium according to Fiorentino et al. [59], with some modifications. In detail, 10 g of root–substrate was suspended in sterile distilled water to provide serial dilutions (four replicates per dilution), and 100 μL aliquots of each sample were spread on the surface of 90 mm culture plates containing Rose Bengal Chloramphenicol agar (Merck KGaA, Darmstadt, Germany). The plates were incubated at 25 °C and examined daily for emerging fungal colonies. The results have been expressed as the CFU per g of dry substrate.

### 4.5. Statistical Analysis

The data were tested for a normal distribution through the Shapiro–Wilk test. Thus, the data were analyzed by a two-way ANOVA (*T. atroviride* treatment and substrate as variables), followed by a Tukey’s post hoc test (*p* < 0.05). The statistical analyses and graphs were processed using Prism 10 (GraphPad Software Inc., La Jolla, CA, USA). Principal component analysis (PCA) was performed using Prism 10, selecting eigenvalues higher than 1.0.

## 5. Conclusions

Coconut coir dust confirmed its suitability as an alternative substrate to peat for bedding plant production. Regarding fungal inoculum applications, *T. atroviride* improved both the quantitative and qualitative parameters of *I. walleriana* bedding plants. The inoculum positively affected the biomass parameters, nutrient uptake, and leaf pigment concentrations without particular differences between peat and coir. *T. atroviride* promoted the aesthetic parameters important for product marketability, such as the number and dimensions of the flowers and the greenness index. In light of our results, the application of coconut coir dust in combination with *Trichoderma* spp. is worthy to be explored in other bedding plants with the aim of supporting growers in the development of sustainable horticultural practices, including peat substrate replacement and use of beneficial microorganisms.

## Figures and Tables

**Figure 1 plants-13-00583-f001:**
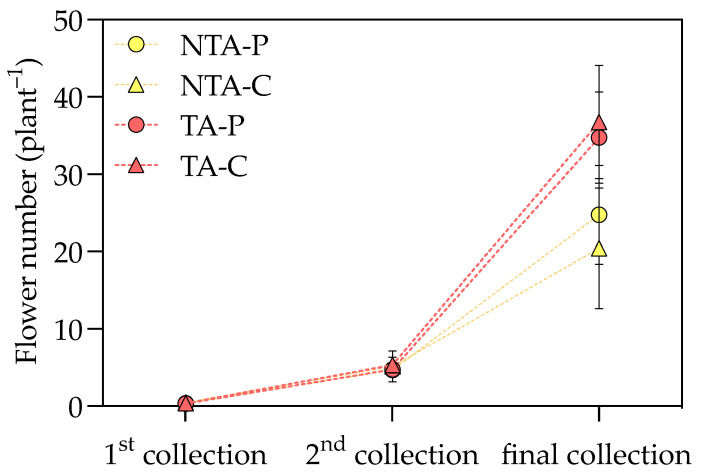
The flower number measured at three time points during the experiment. Values represent the means (*n* = 4) ± SDs as Y-bars. NTA = no *T. atroviride* treatment, TA = *T. atroviride* treatment, P = peat, C = coir. Two-way ANOVA results = Treatment: *p* = ns at the first and second sampling points, *p* < 0.01 at the final sampling point. Substrate: *p* = ns at all the sampling points. Treatment × Substrate: *p* = ns at all the sampling points.

**Figure 2 plants-13-00583-f002:**
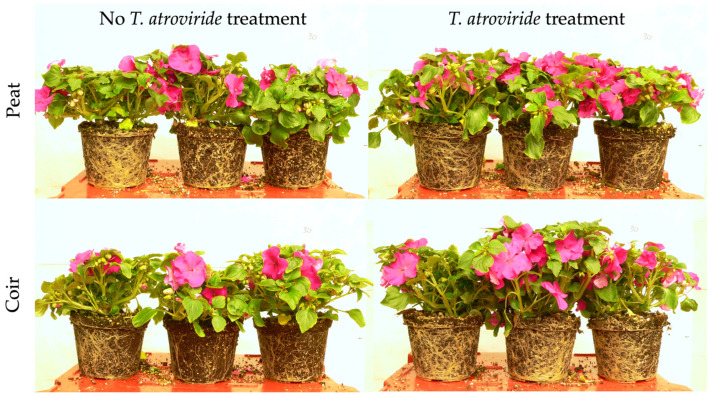
*Impatiens walleriana* bedding plants at the end of the experiment, 56 days after transplant.

**Figure 3 plants-13-00583-f003:**
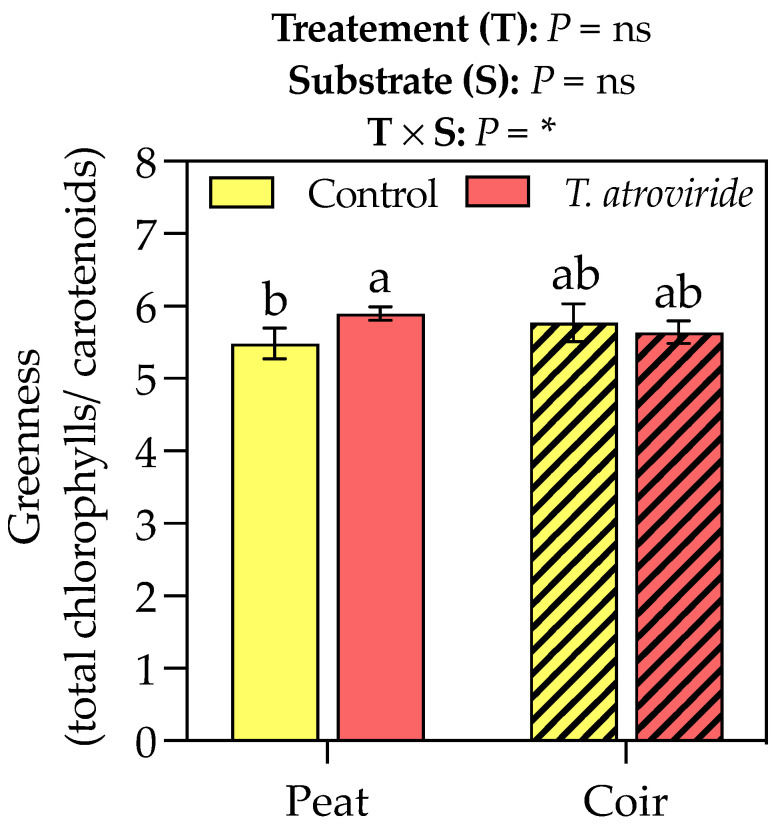
Greenness index (chlorophyll *a* + *b* to total carotenoids ratio) at the end of the experiment. Bars represent the means (*n* = 4) ± SDs. Two-way ANOVA *p*-values and letters corresponding to Tukey’s post hoc results are reported in the figure (* *p* < 0.05; ns = not significant).

**Figure 4 plants-13-00583-f004:**
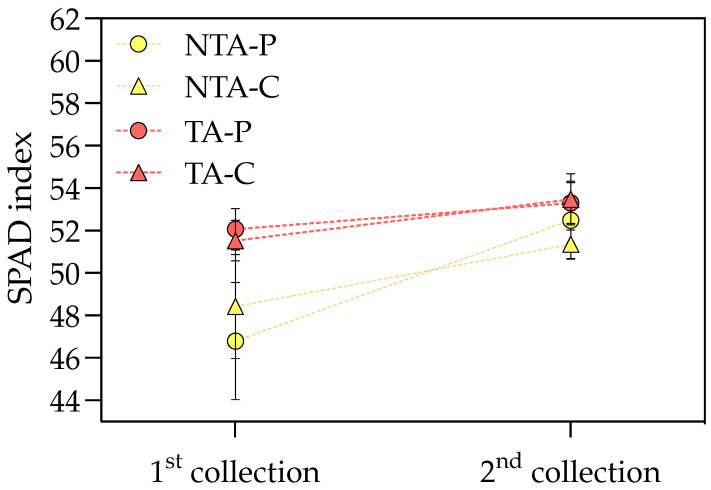
SPAD index at the two intermediate sampling points during the experiment. Values represent the means (*n* = 4) ± SDs as Y-bars. NTA = no *T. atroviride* treatment, TA = *T. atroviride* treatment, P = peat, C = coir. Two-way ANOVA results = Treatment: *p* < 0.01 at the first sampling point, *p* < 0.05 at the second sampling point. Substrate: *p* = ns at both sampling points. Treatment × Substrate: *p* = ns at both sampling points.

**Figure 5 plants-13-00583-f005:**
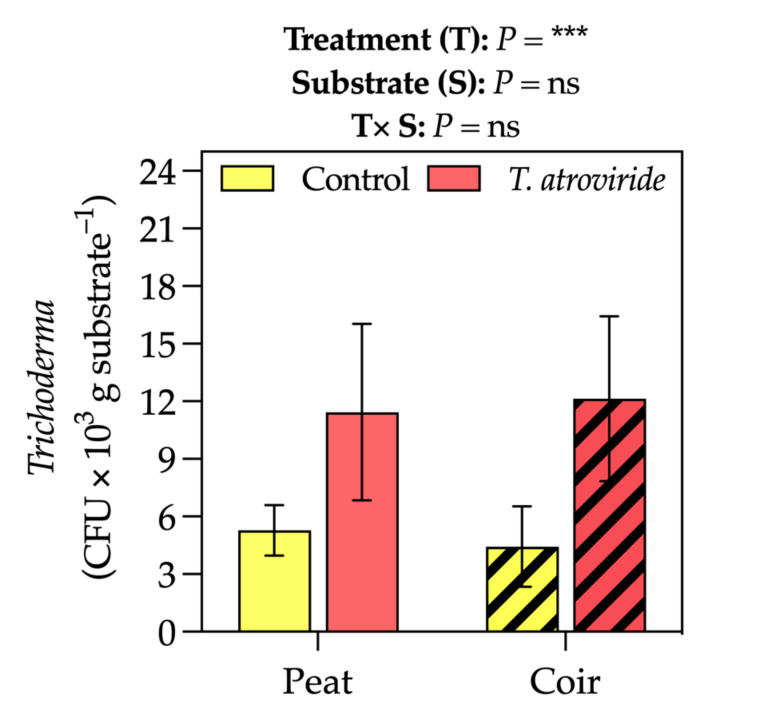
*Trichoderma* colony-forming unit (CFU) in the substrate at the intermediate sampling point. Bars represent the means (*n* = 4) ± SDs. Two-way ANOVA *p*-values and Tukey’s post hoc results are reported in the figure (*** *p* < 0.001; ns = not significant).

**Figure 6 plants-13-00583-f006:**
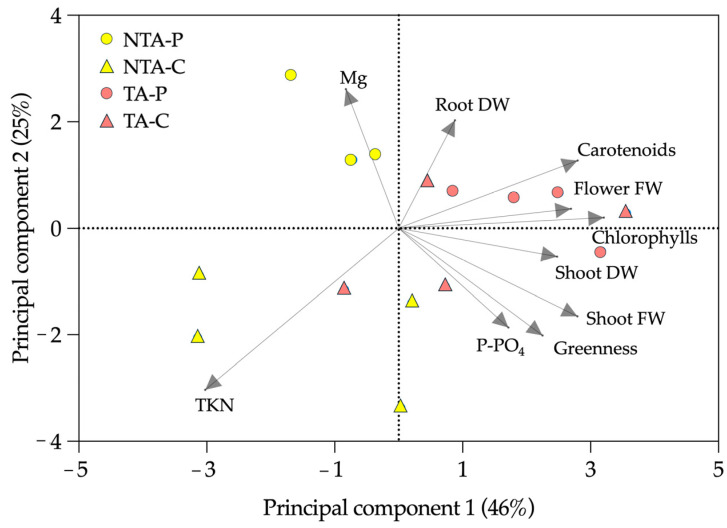
Principal component analysis (PCA) of the experimental results. The loading factors (grey arrows) were selected using eigenvalues higher than 1.0. NTA = no *T. atroviride*, TA = *T. atroviride*, P = peat, C = coir, FW = fresh weight, DW = dry weight, TKN = total Kjeldahl nitrogen.

**Table 1 plants-13-00583-t001:** Biometric parameters of *I. walleriana* plants at the end of the experiment.

Source of Variation	Shoot FW (g plant^−1^)	Shoot DW (g plant^−1^)	Total Flower FW (g plant^−1^)	Total Flower Number (plant^−1^)	Leaf Area (cm^2^ plant^−1^)	Root DW (g plant^−1^)
Treatment (T)						
No *T. atroviride* (NTA)	109 ± 13.7	5.1 ± 0.44	20.2 ± 7.41	28.0 ± 8.21	2850 ± 415.6	1.4 ± 0.20
*T. atroviride* (TA)	135 ± 12.9	5.9 ± 0.77	27.0 ± 2.41	41.2 ± 5.70	3519 ± 343.6	1.6 ± 0.17
*p*-value	**	*	*	**	**	ns
Substrate (S)						
Peat (P)	117 ± 19.4	5.5 ± 0.94	24.4 ± 5.33	34.8 ± 7.92	3098 ± 569.2	1.6 ± 0.16
Coir (C)	126 ± 18.0	5.4 ± 0.51	22.8 ± 7.57	34.3 ± 11.7	3270 ± 453.3	1.4 ± 0.14
*p*-value	ns	ns	ns	ns	ns	**
T × S						
NTA × P	101 ± 9.7	4.9 ± 0.35	22.5 ± 6.96	29.8 ± 6.86	2679 ± 442.4	1.6 ± 0.18
NTA × C	116 ± 14.4	5.2 ± 0.50	17.9 ± 8.10	26.1 ± 10.0	3021 ± 359.8	1.3 ± 0.12
TA × P	133 ± 11.2	6.2 ± 0.90	26.4 ± 2.79	39.9 ± 5.65	3517 ± 302.4	1.7 ± 0.14
TA × C	137 ± 15.9	5.5 ± 0.56	27.7 ± 2.14	42.5 ± 6.28	3520 ± 428.9	1.5 ± 0.13
*p*-value	ns	ns	ns	ns	ns	ns

Values represent the means (*n* = 4) ± SDs. Two-way ANOVA *p*-values are reported in the table (* *p* < 0.05; ** *p* < 0.01; ns = not significant). FW = fresh weight; DW = dry weight.

**Table 2 plants-13-00583-t002:** Nutrient concentrations (g kg^−1^ DW) measured in *I. walleriana* shoots at the end of the experiment.

Source of Variation	TKN	P-PO_4_	K	Ca	Mg
Treatment (T)					
No *T. atroviride* (NTA)	42.6 ± 2.13	7.6 ± 0.50	38.7 ± 2.00	28.5 ± 0.82	5.7 ± 0.51
*T. atroviride* (TA)	40.8 ± 0.99	8.1 ± 0.44	42.1 ± 2.77	30.3 ± 0.76	5.6 ± 0.28
*p*-value	**	*	*	**	ns
Substrate (S)					
Peat (P)	40.9 ± 1.07	7.7 ± 0.61	39.2 ± 2.48	29.4 ± 0.67	5.9 ± 0.32
Coir (C)	42.6 ± 2.16	8.0 ± 0.41	41.6 ± 2.92	29.5 ± 1.68	5.4 ± 0.28
*p*-value	*	ns	ns	ns	*
T × S					
NTA × P	41.0 ± 1.56 b	7.3 ± 0.36	37.5 ± 1.48	28.9 ± 0.64 bc	6.1 ± 0.30
NTA × C	44.3 ± 0.80 a	7.9 ± 0.47	40.0 ± 1.63	28.1 ± 0.85 c	5.3 ± 0.30
TA × P	40.8 ± 0.48 b	8.1 ± 0.56	40.9 ± 2.03	29.8 ± 0.36 ab	5.7 ± 0.23
TA × C	40.8 ± 1.44 b	8.2 ± 0.38	43.2 ± 3.34	30.8 ± 0.71 a	5.5 ± 0.30
*p*-value	*	ns	ns	*	ns

Values represent the means (*n* = 4) ± SDs. Two-way ANOVA *p*-values and letters are reported in the table (* *p* < 0.05; ** *p* < 0.01; ns = not significant). DW = dry weight; TKN = total Kjeldahl nitrogen.

**Table 3 plants-13-00583-t003:** Pigment concentrations (mg g^−1^ FW) measured in *I. walleriana* leaves at the end of the experiment.

Source of Variation	Chlorophyll *a*	Chlorophyll *b*	Carotenoids
Treatment (T)			
No *T. atroviride* (NTA)	0.97 ± 0.054	0.32 ± 0.024	0.23 ± 0.009
*T. atroviride* (TA)	1.06 ± 0.068	0.34 ± 0.029	0.24 ± 0.013
*p*-value	*	ns	*
Substrate (S)			
Peat (P)	1.03 ± 0.074	0.34 ± 0.033	0.24 ± 0.009
Coir (C)	1.00 ± 0.077	0.32 ± 0.026	0.23 ± 0.015
*p*-value	ns	ns	ns
T × S			
NTA × P	0.98 ± 0.059	0.31 ± 0.028	0.24 ± 0.007
NTA × C	0.97 ± 0.057	0.32 ± 0.022	0.22 ± 0.007
TA × P	1.09 ± 0.035	0.35 ± 0.007	0.25 ± 0.008
TA × C	1.03 ± 0.088	0.33 ± 0.034	0.24 ± 0.017
*p*-value	ns	ns	ns

Values represent the means (*n* = 4) ± SDs. Two-way ANOVA *p*-values are reported in the table (* *p* < 0.05; ns = not significant). FW = fresh weight.

**Table 4 plants-13-00583-t004:** Main chemical–physical characteristics of the two used growing media.

Parameters	Peat:Perlite (70:30 *v v*^−1^)	Coconut Coir Dust:Perlite (70:30 *v v*^−1^)
pH	5.7	7.7
EC (µS cm^−1^)	191.2	136.7
BD (g cm^−3^)	0.129	0.112
TP (% by volume)	92.0	93.8
AWC (% by volume)	27.8	21.7
W-1 kPa (% by volume)	55.4	55.8
AC-1 kPa (% by volume)	19.4	18.6

EC = Electrical conductivity; BD = Bulk density; TP = Total porosity; AWC = Available water content; W-1 kPa = Water-holding capacity; AC-1 kPa = Air content.

## Data Availability

Data are contained within the article or Appendix A.

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
