# Peer review of "Trichoderma atroviride Enhances Impatiens walleriana Hook. f Growth and Flowering in Different Growing Media"

_plants, 2024, doi:10.3390/plants13050583_

Round 1

Reviewer 1 Report

Comments and Suggestions for Authors

The manuscript is related to the Trichoderma significance in Impatiens walleriana bedding plant life by the use of coconut coir as the peat substitute.

If the Trichoderma should be in italics in whole manuscript you correct it..

What do you mean ,,greenness index” and SPAD? Clear.

,,Variance” is not the same as ,,significance”. Correct the statistic in whole manuscript if needed including tables and figures. Error bars are problematic, e.g. Figures 3-5.

Suppl. Tables – the units should be corrected. Root/Shoot? Clear.

The tables are constructed unclears. What is significant in interaction with?

4. Methods – correct the place of paragraph

,,Growing media pH and electrical conductivity (EC) 226 were determined by EN 13037/1999 and EN 13038/1999 methods, …” – the methods should be described.

The content of chemicals and biologically active components in leaves and shoots are significantly diverse. How do you measure it? Why this is the same in work? Precise. This probably is false results.

Conclusions should be rethinking.

Comments on the Quality of English Language

ok.

Author Response

If the Trichoderma should be in italics in whole manuscript you correct it..

Response:

Thank you for the comment, we modified the font in italics throughout the manuscript.

What do you mean ,,greenness index” and SPAD? Clear.

Response:

Thank you for the comment. Greenness index is a direct measurement of leaf pigments (ratio between total chlorophylls and carotenoids, measured by spectrophotometer as described in Material and Methods). SPAD index is an indirect measurement of leaf pigment resulting from the non-destructive determination of absorbance of leaf tissue at two wavelengths. We have used the SPAD index to have intermediate sampling points for pigment analysis, avoiding the removal of leaves, while we have directly measured the pigment concentration at the end of the trial. We further clarified this aspect in Material and Methods.

Variance” is not the same as ,,significance”. Correct the statistic in whole manuscript if needed including tables and figures.

Response:

Thank you for the comment. We changed “source of variance” with “source of variation” throughout the manuscript.

Error bars are problematic, e.g. Figures 3-5.

Response:

We have now reported the standard deviation as above and below bars to be clearer. However, if this is not enough, could you please kindly better explain this request?

Suppl. Tables – the units should be corrected. Root/Shoot? Clear.

Response:

Thank you for the comment. The Root/Shoot ratio did not have a unit of measure because it is just a ratio.

The tables are constructed unclears. What is significant in interaction with?

Response:

We have now included P-value line text, changed “inoculum” with “treatment”, as well as we have inserted the Trichoderma species. However, if this is not enough, could you please kindly better explain this request?

  1. Methods – correct the place of paragraph

,,Growing media pH and electrical conductivity (EC) 226 were determined by EN 13037/1999 and EN 13038/1999 methods, …” – the methods should be described.

Response:

Thank you for the comment, we have added a brief description of methods.

The content of chemicals and biologically active components in leaves and shoots are significantly diverse. How do you measure it? Why this is the same in work? Precise. This probably is false results.

Response:

Thank you for the comment. The nutrient and chlorophyll contents of Impatiens shoots and leaves have been updated for better clarity. Section 4.3 details the methods used to measure the concentrations of nutrients in Impatiens shoots (leaves plus stem), as well as the concentrations of pigments in Impatiens leaves clearly indicating the type of sample used for the analysis. Table 2 now specifies that the reported nutrient contents (expressed as g kg DW-1) were measured in Impatiens shoots, while the values in Table 3 (expressed as mg g FW-1) refer to the chlorophyll and carotenoid contents of Impatiens leaves.

These measurements, collectively, suggest that Impatiens plants in Trichoderma-inoculated growing media leads to increased nutrient absorption efficiency, which results in lower nitrogen requirements and higher concentrations of phosphate, calcium, and potassium. In addition, it leads to greater chlorophyll and carotenoid contents in Impatiens leaves.

Conclusions should be rethinking.

Response:

Thank you for the comment. We have added some future perspectives in conclusions.

Reviewer 2 Report

Comments and Suggestions for Authors

Review

Trichoderma enhances Impatiens walleriana growth and flowering in different growing media

Your research is not something new, it is known. It's good that you did it on a flower species Impatiens  that is popular with flower growers though. Also, growing mediums are common in the production of vegetables and flowers. The use of Trichoderma fungus is also something normal in today's horticulture production.

Results stated that Trichoderma can readily colonize coir, and the same positive effects of inoculum were found in plants grown on both substrates. The biostimulant effect of Trichoderma was retrieved as an increase in aboveground biomass, number and weight of flowers, pigments and nutrient concentration thereby improving the commercial quality of Impatiens. Thus, Trichoderma has shown its potential in making bedding plant cultivation more sustainable, improving the yield and aesthetic parameters of plants grown on peat and coconut coir dust substrates.

Your research was on the influence of the fungus T.a. on the growth and development of the flowering plant species Impatiens walleriana on three different substrate media. That is OK. What is not correct is that, and you stated that the role of T.a. biostimulator. Not. Trichoderma atroviride is a mycoparasite saprophyte, meaning it parasitizes other fungi and it obtains its food by absorbing dissolved organic matter. Trichoderma atroviride are found in many substrates including decaying wood, and fungal fruiting bodies, and are known to be able to use a wide range of substrates for carbon and nitrogen sources. Trichoderma atroviride has many uses that make them relevant to humans. They are used as natural biocontrol agents for both insects and other fungi.

Trichoderma is not a biostimulator but has a bio control......

I kindly ask the authors to think about that sentence and it would be good if they replaced that meaning.

Author Response

Your research is not something new, it is known. It's good that you did it on a flower species Impatiens that is popular with flower growers though. Also, growing mediums are common in the production of vegetables and flowers. The use of Trichoderma fungus is also something normal in today's horticulture production.

Results stated that Trichoderma can readily colonize coir, and the same positive effects of inoculum were found in plants grown on both substrates. The biostimulant effect of Trichoderma was retrieved as an increase in aboveground biomass, number and weight of flowers, pigments and nutrient concentration thereby improving the commercial quality of Impatiens. Thus, Trichoderma has shown its potential in making bedding plant cultivation more sustainable, improving the yield and aesthetic parameters of plants grown on peat and coconut coir dust substrates.

Your research was on the influence of the fungus T.a. on the growth and development of the flowering plant species Impatiens walleriana on three different substrate media. That is OK. What is not correct is that, and you stated that the role of T.a. biostimulator. Not. Trichoderma atroviride is a mycoparasite saprophyte, meaning it parasitizes other fungi and it obtains its food by absorbing dissolved organic matter. Trichoderma atroviride are found in many substrates including decaying wood, and fungal fruiting bodies, and are known to be able to use a wide range of substrates for carbon and nitrogen sources. Trichoderma atroviride has many uses that make them relevant to humans. They are used as natural biocontrol agents for both insects and other fungi.

Trichoderma is not a biostimulator but has a bio control......

I kindly ask the authors to think about that sentence and it would be good if they replaced that meaning.

Response:

Trichoderma species are not included in EU Fertilizer Regulation 2019/1009 (Component Material Categories, number 7) (CMC-7) as microbial plant biostimulants. They are actually registered as microbial biological control agent, as a biopesticide against plant pathogens. However, different species of Trichoderma are found frequently as active ingredients in many commercial formulations with indications as biofertilizers, bio-growth enhancers, and biostimulants due to the ability of these biological components to promote plant growth, abiotic stress tolerance, improve yield and nutritional quality, as proven in various crop studies. In particular, Trichoderma atroviride is recognized to have a powerful capacity to improve plant biomass of different crops (10.17660/ActaHortic.2020.1268.26, 10.1111/j.1439-0329.2010.00710.x, 10.3390/plants11091175, 10.1111/j.1364-3703.2010.00674.x, 10.1002/jsfa.6875, 10.1111/pce.14014, 10.1094/MPMI-01-15-0005-R.).

We agree with the relevance of this punctuation and have therefore highlighted these aspects in the introduction to the manuscript.

Reviewer 3 Report

Comments and Suggestions for Authors

The article submitted for review covers research topics that have been pursued by numerous scientific and research institutions for a long time. In the research, the authors used a biostimulator whose active substance is Trichoderma atroviride, but apart from the methodology, throughout the article they only use the generic name, which is a serious omission and should be corrected. I have the impression that the Authors are not specialists in microorganisms, hence numerous errors in vocabulary.

I believe that after editing, taking into account the comments and clarifying the issues I provide below, you can consider accepting the article for the Plants journal.

Comments:

Line 2 - Be sure to add in the title - "Trichderma atroviride" and "Impatiens walleriana Hook. f” - after all, it is a scientific work, not a popular science magazine or a guide for allotment gardeners.

Line 24- Be sure to write the species name Trichoderma atroviride. Consistently use T. atroviride in the abstract.

Line 35 – replace "beneficial fungi" with Trichoderma atroviride

Line 63 – What bacteria? Please list their names.

Line 64 - Please expand your knowledge about these microorganisms (what genera, families, species do they represent). This information is too general.

Lines 68-69 - Please expand on the topic of Trichoderma mushrooms, which species are used, according to the previous comment.

Line 70 - Please add scientific information, because the information presented in the article sounds like something from a handbook for gun owners.

Line 75 – which Trichoderma? T.atroviride?

Line 82 and provide the full name of the fungus T. atroviride throughout the article.

Lines 90-91 What did this mean? perhaps the multiplication of the inoculum in the substrate was the reason for such results. Fungi of the genus Trichoderma are characterized by rapid growth of mycelium.

Line 97 - there cannot be "Trichoderma inoculum", only inoculation with Trichoderma fungus (inoculum - the amount of infectious material)

Table 2 - Why is the entry Kjeldahl N in table 2?

Line 127 - "Trichoderma inoculum" should be as before in line 97 (the inoculum is constantly multiplying, and in unfavorable conditions it may die).

Line 129 - inoculum - note as above!

Line 130 - interaction between inoculum and substrate - this interaction should be looked at (did the substrate stimulate the development of the fungus?)

line 167 – it is "Trichoderma inoculum" - it should be "inoculation with Trichoderma fungus"

or "inoculation of the substrate with Trichoderma fungus". Specifically which species of Trichoderma?

Line 177 - what strain of Trichoderma?

Line 178 - what Trichoderma - please tell me the species.

Line 180 – as above, specify the species Trichoderma

Line 185 – what Trichoderma?

Line 187 - there is "Trichoderma inoculum" and it should be T. atroviride, without inoculum

Lines 190 – 205 – written too generally, what does “significantly” mean? something has "significantly" increased, "significantly improved", etc. - specifically, if the research effects are expressed in numbers, they should be used. I would like to remind you once again that you cannot generalize when writing the generic name Trichoderma. Many authors studied various species, e.g. T.viride, T.harzianum, T.koningii, T.hamatum, T. asperellum, T.reesei and T.atroviride.

If the authors had carefully studied the literature, they certainly would not have made these mistakes. Once again, everything cannot be reduced to the broad generalization of TRICHODERMA. Please definitely improve the discussion.

Line 2011 – move one line down “4. "Materials and Methods"

Lines 221-226 – Please write clearly how many combinations there were and how many repetitions? Is 1 pot treated as a repetition? According to me, there were 4 combinations with 80 repetitions in each (pots-plants)

Line 243 – were analyzes performed on all plants, or maybe only a few?

Line 246- how many leaves from the combination were measured?

line 253 – 10 plants from a repetition or a combination?

line 256 - what does "(100 mg FW per replicate)" mean - explain as before what is a repetition? How many repetitions are there? My guess is 80 reps for each combination.

282- it is better to use the term biostimulator, Trichoderma atroviride, rather than inoculum.

Line 288 – It is necessary to use the name of the species, i.e. Trichoderma atroviride

Line 289 - you cannot write like this: "inoculum" did not affect - the T.atroviride fungus or biostimulator did.

Line 291 - "Trichoderma promoted" - you absolutely cannot write like that - specifically T.atroviride as a component of the culture medium..., or a biostimulator based on T.atroviride....

Author Response

Line 2 - Be sure to add in the title - "Trichderma atroviride" and "Impatiens walleriana Hook. f” - after all, it is a scientific work, not a popular science magazine or a guide for allotment gardeners.

Response:

We have added the scientific names as suggested.

Line 24- Be sure to write the species name Trichoderma atroviride. Consistently use T. atroviride in the abstract.

Response:

Thank you for the comment, we modified accordingly throughout the manuscript.

Line 35 – replace "beneficial fungi" with Trichoderma atroviride

Response:

We have replaced the keyword “beneficial fungi” with “Trichoderma spp.” to avoid repetition of words already used in the title.

Line 63 – What bacteria? Please list their names.

Response:

Thank you for the comment. In the cited reference the species of bacteria have not been reported. However, we have now reported the bacterial functions important for promoting plant growth (e.g., phosphate solubilization) and we have also added another reference.

Line 64 - Please expand your knowledge about these microorganisms (what genera, families, species do they represent). This information is too general.

Response:

We have added more information about fungal species throughout the manuscript.

Lines 68-69 - Please expand on the topic of Trichoderma mushrooms, which species are used, according to the previous comment.

Response:

Thank you for the comment. We have added several references about Trichoderma species and T. atroviride (References 23, 27, 28, 29, 30, 31, 32, 33, 34, 35).

Line 70 - Please add scientific information, because the information presented in the article sounds like something from a handbook for gun owners.

Response:

We have improved the use of scientific names.

Line 75 – which Trichoderma? T.atroviride?

Response:

We have now always included the Trichoderma species.

Line 82 and provide the full name of the fungus T. atroviride throughout the article.

Response:

We have modified Trichoderma with T. atroviride along the manuscript.

Lines 90-91 What did this mean? perhaps the multiplication of the inoculum in the substrate was the reason for such results. Fungi of the genus Trichoderma are characterized by rapid growth of mycelium.

Response:

Despite the rapid expected growth of Trichoderma, as also highlighted by results presented in Figure 5, we did not observe differences in the first and second flower collections. This might be due to the also rapid cycle of I. walleriana.

Line 97 - there cannot be "Trichoderma inoculum", only inoculation with Trichoderma fungus (inoculum - the amount of infectious material)

Response:

Thank you for the comment, we changed “Trichoderma inoculum” with “T. atroviride treatment” throughout the manuscript.

Table 2 - Why is the entry Kjeldahl N in table 2?

Response:

We have now used the abbreviation TKN to indicate the total Kjeldahl nitrogen.

Line 127 - "Trichoderma inoculum" should be as before in line 97 (the inoculum is constantly multiplying, and in unfavorable conditions it may die).

Response:

We changed “Trichoderma inoculum” with “T. atroviride treatment” throughout the manuscript.

Line 129 - inoculum - note as above!

Response:

We changed “Trichoderma inoculum” with “T. atroviride treatment” throughout the manuscript.

Line 130 - interaction between inoculum and substrate - this interaction should be looked at (did the substrate stimulate the development of the fungus?)

Response:

Regarding the greenness index, the interaction is referred to the effect of both factors on plants. However, the effect of substrate in stimulating the Trichoderma colonization is reported in Figure 5.

line 167 – it is "Trichoderma inoculum" - it should be "inoculation with Trichoderma fungus"

Response:

We changed “Trichoderma inoculum” with “T. atroviride treatment” throughout the manuscript.

or "inoculation of the substrate with Trichoderma fungus". Specifically which species of Trichoderma?

Response:

Thank you for the comment, we modified the sentence accordingly.

Line 177 - what strain of Trichoderma?

Response:

We have specified the species, i.e., T. harzianum.

Line 178 - what Trichoderma - please tell me the species.

Response:

Since this is something common for several species of Trichoderma, we have modified the term in Trichoderma spp.

Line 180 – as above, specify the species Trichoderma.

Response:

We have specified the species, i.e., T. virens.

Line 185 – what Trichoderma?

Response:

Since this is something common for several species of Trichoderma, we have modified the term in Trichoderma spp.

Line 187 - there is "Trichoderma inoculum" and it should be T. atroviride, without inoculum

Response:

We modified accordingly.

Lines 190 – 205 – written too generally, what does “significantly” mean? something has "significantly" increased, "significantly improved", etc. - specifically, if the research effects are expressed in numbers, they should be used. I would like to remind you once again that you cannot generalize when writing the generic name Trichoderma. Many authors studied various species, e.g. T.viride, T.harzianum, T.koningii, T.hamatum, T. asperellum, T.reesei and T.atroviride.

Response:

We added Trichoderma species along the manuscript.

If the authors had carefully studied the literature, they certainly would not have made these mistakes. Once again, everything cannot be reduced to the broad generalization of TRICHODERMA. Please definitely improve the discussion.

Response:

We added Trichoderma species along the manuscript.

Line 2011 – move one line down “4. "Materials and Methods"

Response:

We have modified accordingly.

Lines 221-226 – Please write clearly how many combinations there were and how many repetitions? Is 1 pot treated as a repetition? According to me, there were 4 combinations with 80 repetitions in each (pots-plants)

Response:

Thank you for the comment. Each combination (i.e., treatment) had 4 replicates. Each replicate consists of 20 plants to assure an adequate representativeness. Thus, 80 plants were the total number of plants belonging to the same treatment.

Line 243 – were analyzes performed on all plants, or maybe only a few?

Response:

Thank you for the comment. All the measurements were done on 10 plants out of 20 constituting a replicate. Analyses of biomass, pigments, and nutrients were performed on a bulk sample that brought together the 10 plants. The flower intermediate collections were performed on all plants constituting a replicate (20 plants). We further clarified these aspects in Material and Methods.

Line 246- how many leaves from the combination were measured?

Response:

The leaf area was measured on about 40 g of fresh leaves (on average a 10% of the total biomass).

line 253 – 10 plants from a repetition or a combination?

Response:

From each replicate, 40 plants in total (i.e., 10 plant per replicate, 4 replicates per combination).

line 256 - what does "(100 mg FW per replicate)" mean - explain as before what is a repetition? How many repetitions are there? My guess is 80 reps for each combination.

Response:

Each replicate constited of leaf disks randomly collected from the 10 plants from which it was composed. Thus, a total of 4 replicates per each combination were analyzed in two technical replicates (32 measurements in total). We further clarified this aspect.

282- it is better to use the term biostimulator, Trichoderma atroviride, rather than inoculum.

Response:

We changed “Trichoderma inoculum” with “T. atroviride treatment” throughout the manuscript.

Line 288 – It is necessary to use the name of the species, i.e. Trichoderma atroviride

Response:

We have modified accordingly.

Line 289 - you cannot write like this: "inoculum" did not affect - the T.atroviride fungus or biostimulator did.

Response:

We changed “Trichoderma inoculum” with “T. atroviride treatment” throughout the manuscript.

Line 291 - "Trichoderma promoted" - you absolutely cannot write like that - specifically T.atroviride as a component of the culture medium..., or a biostimulator based on T.atroviride....

Response:

We changed “Trichoderma inoculum” with “T. atroviride treatment” throughout the manuscript.

Reviewer 4 Report

Comments and Suggestions for Authors

I have suggested some changes in the attached PDF, which are self-explanatory.

Author Response

I have suggested some changes in the attached PDF, which are self-explanatory.

Response:

We would like to thank the Reviewer; we reported our point-by-point answer to all queries.

What was the ratio of each substrate?

Response:

The substrate ratio was reported in Material and Methods. However, we have now added the ratio also in the abstract, i.e., 70:30.

Is it the latest available data?

Response:

Thank you for the comment. Yes, it is the last report of European Commission on flower and ornamental plants statistics.

Please include few more references indicating the usefulness on the coir application with reference to Impatiens or other ornamentals.

Response:

Thank you for the comment. We have added two more references indicating the use of coir on bedding plants such as Impatiens spp. (References 14 and 15).

Please include few more references indicating the usefulness on the Trichoderma spp. application with reference to Impatiens or other ornamentals.

Response:

Thank you for the comment. We have added several references about Trichoderma species and T. atroviride (References 23, 27, 28, 29, 30, 31, 32, 33, 34, 35).

long flower display, attractive leaves

Response:

We have modified accordingly.

27 g, check Table 1

Response:

Thank you for the comment, the error was in the table, not inoculated shoot FW was 109 rather than 108 (108.68 rounded to 109). We corrected the table.

That should be in the Table's footnote.

Response:

We moved the description in the caption in all tables.

standard deviation (as Y-bars)

Response:

We have included in the text “(as Y-bars)” as suggested.

We have also modified the italic font for Thricoderma and Impatiens throughout the manuscript.

Round 2

Reviewer 1 Report

Comments and Suggestions for Authors

The manuscript is corrected satisfactory.

Comments on the Quality of English Language

The language of the work is generally correctly.

Reviewer 3 Report

Comments and Suggestions for Authors

I accept the revised version of the manuscript.